# OpenReview forum: "A Critical Look At Tokenwise Reward-Guided Text Generation"
_colmweb.org/COLM/2025/Conference — COLM 2025_

### Official Review · Reviewer_JW6d · 2025-04-17

**Rating:** 8
**Confidence:** 3
**Ethics Flag:** 1

**Summary:**

This paper addresses a significant problem in the field of reward-guided text generation. The use of reward models trained on full sequences to guide token-by-token generation, where the authors show can be problematic.

1. A theoretical analysis shows why the reward models trained on full sequences can assign arbitrary rewards to partial sequences, leading to suboptimal guidance during decoding.
2. It provided a novel solution namely PARGS (Partial Alignment as Reward-Guided Sampling) to train Bradley-Terry reward models on the partial sequences.
3. It gave the theoretical proof that the sampling policy representing a ratio of two distinct RLHF policies. It is a necessary trade-off to make tokenwise RGTG both principled and tractable.
4. It also made extensive experiments on multiple datasets (summarization, dialogue, text generation). It shows that PARGS outperforms the existing RGTG methods. In addition, it is competitive with more expensive fine-tuning approaches like PPO and DPO.

The paper is well-structured and clearly written. The theoretical contributions are sound and well-supported by proofs. The empirical evaluation is comprehensive, covering multiple tasks, models, and baselines. The approach is practical and addresses an important limitation of existing methods without requiring expensive LLM fine-tuning.

**Questions To Authors:**

None

**Reasons To Accept:**

1. The work provided theoretical analysis on the proposed RGTG methods using full-sequence reward models for partial-sequence scoring on why it can be problematic.
2. The empirical results show the PARGS outperforms the existing RGTG methods, e.g., on multiple tasks. It also performs than more expensive fine-tuning approaches, e.g., PPO and DPO, etc.
3. It enables better alignment even without requiring expensive LLM fine-tuning.

**Reasons To Reject:**

1. The work proposal introduced significant computational overhead during the inference, since the requirement of multiple forward passes through a reward model for each token generation step.
2. The proposed approach assumes preferences are over full sequences transfer to partial sequences. It may not hold in practicealways. Although the paper has acknowledged this limitation, but it does NOT explore it.
3. As the authors mentioned in the limitations section, the proposed approach maybe NOT suitable for the tasks which require step-wise rewards rather than token-wise rewards (e.g., mathematical reasoning).

---

> ### Author Response · Authors · 2025-06-03
>
> Thank you for your positive review.

---

> > ### Comment · Reviewer_JW6d · 2025-06-10
> >
> > Thanks!

---

### Official Review · Reviewer_Z2q2 · 2025-04-25

**Rating:** 6
**Confidence:** 4
**Ethics Flag:** 1

**Summary:**

This paper examines and improves token-wise reward-guided text generation methods for LLMs. It proposes a new approach called PARGS to address issues with existing RGTG methods that use reward models trained on full sequences to score partial sequences during decoding. The authors show that such reward models can assign arbitrary scores to partial sequences, leading to sub-optimal performance. To fix this, they train a Bradley-Terry reward model explicitly on partial sequences and sample from the implied token-wise policy during decoding. This approach outperforms previous RGTG methods like ARGS and CD, achieving results comparable to expensive offline RLHF methods like PPO and DPO without large-scale LLM fine-tuning.

**Questions To Authors:**

- In Table 1, why is the PPO result lacking on the dialogue task?
- Is this issue related to training a PRM? Can the proposed method be applied to train a better PRM?

**Reasons To Accept:**

- This problem, which involves using a reward model trained at the sequence level to improve token-wise alignment, is interesting. To my knowledge, this is the first work to discuss this issue.

- The proposed method appears to be simple yet effective. We do not need to modify much code to implement it successfully.

**Reasons To Reject:**

- One primary concern lies in Eq. 5, which simply discards part of the sequence, specifically y1 and y2. This raises the question of whether we might lose valuable information, potentially introducing bias. While the paper provides some theoretical motivation for this approach, I remain somewhat concerned about this aspect. For example, there might be a case where y2 performs better than y1 in the earlier part of the sequence, but due to some errors that occur in the sampling process of the latter part, y2 ends up being worse than y1 overall. In such a scenario, can we confidently claim that y1 is always superior to y2 in the preceding segment?

- Another concern is the insufficient number of experiments. While summarization and dialogue are common settings, there is a growing trend towards using reward-based optimization to enhance reasoning. Therefore, the paper could benefit from including experiments related to reasoning.

---

> ### Author Response · Authors · 2025-06-03
>
> Thank you for your review and comments.
>
> ## Lemma 2
>
> Given two partial sequences A and B the one with the best completion should be preferred. Our reward model is trained to prefer the winning sequence from the preference data. Therefore, as long as the winning sequence is close to optimal or human generated the algorithm would do the correct thing.
>
> ## Experiments
>
> Our goal was to conduct comprehensive experiments on these domains, summarization and dialogue, since all the RGTG baselines also presented experiments on these domains. We presented results on multiple datasets and language models and evaluated on multiple metrics. We present multiple studies in the Appendix including significance testing (Appendix D) and human evaluation (Appendix H). Additionally, we present results on machine translation and diversity (Appendix F). Across all these, we demonstrate the strength of our method PARGS.
>
> Reasoning is an important domain and we leave it for future work.
>
> ## PPO on dialogue
>
> The dialogue experiments were on 7 billion language models and the PPO experiments required too much compute. For these experiments we presented DPO as the RLHF baseline.
>
> ## Process Reward Model
>
> The proposed method improves sampling from the language model. Although, it is orthogonal to training a PRM, a better sampling algorithm can reduce the reliance on a PRM.

---

> > ### Comment · Reviewer_Z2q2 · 2025-06-04
> >
> > Thank you for your response and the clarification regarding the proof of Lemma 2. I have reviewed it and understand the derivation.
> > However, I'm still finding it challenging to see how this proof fully addresses my initial concern (R1/W1). Could you perhaps elaborate further on the connection or provide additional details that specifically target the R1/W1 point?
> >
> > >One primary concern lies in Eq. 5, which simply discards part of the sequence, specifically y1 and y2. This raises the question of whether we might lose valuable information, potentially introducing bias. While the paper provides some theoretical motivation for this approach, I remain somewhat concerned about this aspect. For example, there might be a case where y2 performs better than y1 in the earlier part of the sequence, but due to some errors that occur in the sampling process of the latter part, y2 ends up being worse than y1 overall. In such a scenario, can we confidently claim that y1 is always superior to y2 in the preceding segment?

---

> > > ### Author Response · Authors · 2025-06-10
> > >
> > > Thank you for the follow-up question. Let's take two full sequences $y_1$ and $y_2$ and suppose that $y_1$ is the winning sequence. Now let $y_1^k$ and $y_2^k$ be two k-th length subsequences. $y_2^k$ can be a winning sequence if there is a continuation of $y_2^k$ that is better than $y_1$. We cannot guarantee that that scenario will never happen. In order to determine that we either need multiple calls to a language model for multiple completions and a way to evaluate them or a way to annotate subsequences that is difficult for human annotators. But we argue that our assumption is reasonable and backed by strong empirical results.

---

> > > > ### Comment · Reviewer_Z2q2 · 2025-06-10
> > > >
> > > > Thank you for your response. I agree that this issue likely exists and may warrant further analysis or at least some discussion. I understand that this is not something that can be fully addressed within just a few days. I have raised my score, and I hope that future work will be able to resolve this issue.

---

> > > > > ### Author Response · Authors · 2025-06-11
> > > > >
> > > > > Thank You.

---

### Official Review · Reviewer_2dhF · 2025-04-29

**Rating:** 6
**Confidence:** 5
**Ethics Flag:** 1

**Summary:**

This work theoretically and empirically analyzes Tokenwise Reward-Guided Text Generation (RGTG) methods, where a reward model predicts the marginal token probability that is added to the original token probability of the frozen base LLM to approximate an RLHF-tuned policy. Existing works in RGTG overlooked the credit assignment problem, failing to compute partial token-level rewards properly. The paper proposes a simple remedy of training the reward model with *all* prefix pairs of the winning/losing sequences (dubbed PARGS), ensuring that the final reward is properly distilled to each token. Empirical studies on different tasks (summarization, helpful-harmful dialogue, UltraFeedback win/lose data) show the effectiveness of the proposed method.

**Questions To Authors:**

(1) (2) See Reasons to Reject, point 1.

(3) **Directly measuring credit assignment**: One interesting method for measuring the token-level credit assignment ability is token mean reciprocal rank, introduced in [1] (Section 5.2). In this approach, you sample multiple sequences (>=16), determine one with the highest reward (gold), and check the reciprocal rank of each token in the gold sequence under the manipulated distribution. The mean reciprocal rank closer to 1 will imply that the tokens with higher final reward have been assigned a higher probability. Can PARGS achieve a higher token mean reciprocal rank than other RGTG methods? (This is not necessary; given the limited time, I would be happy to increase my score when (1) and (2) are fully addressed.)

[1] Lee, Y., Lee, J., & Hwang, S. W. (2023, December). Learning to Rank Generation with Pairwise Partial Rewards. In Proceedings of the 2023 Conference on Empirical Methods in Natural Language Processing (pp. 6078-6092).

Note: After checking the author's response, I increased the confidence from 3 to 5, as I am now confident that the core assumption in this paper's theoretical analysis is severely unjustified, which limits the overall contribution.

**Reasons To Accept:**

1. **Clear theoretical analysis of existing RGTG works.** The authors provide convincing theoretical analyses about how existing RGTG works overlooked the credit assignment problem when distilling sequence-level rewards to token-level rewards. Based on the results, the authors successfully derived the novel training loss (Eqn (4)) and decoding algorithm (Theorem 3).

2. **Diverse experiments that back the theoretical analyses.** Experimental results (e.g., average rewards and LLM-as-a-judge) clearly show that the reward model trained with the proposed loss produces improved results than baselines that ignore the credit assignment. The fact that the performance of PARGS almost matches RLHF-trained method is quite promising, and opens up many different future directions (e.g., having small specialized token-level reward models specialized in different tasks for plug-and-play architecture, which is a more lightweight solution than plug-and-playing LoRA weights in terms of both training and inference).

**Reasons To Reject:**

1. **Missing empirical justification for a controversial assumption.** The proposed loss function for training a reward model relies on the assumption *"that partial sequences inherit the winning/losing label of full sequences"* (Lines 174-175). I question the validity of such an assumption and believe that the paper requires more empirical justification. I have two questions:

(1) **Does this assumption empirically hold?** The standard, unbiased, and correct approach for estimating the token-wise reward is to obtain the expected final reward by sampling multiple rollouts (e.g., MCTS). A straightforward way to prove that this assumption is correct is to show that the MCTS-calculated reward of the prefix of a winning sequence is actually higher than that of a losing sequence for all prefix lengths. While it is hard to conduct such an experiment in the preference domain, using tasks with easily verifiable rewards (see point 2) will enable such analyses.

(2) **Does the token-level reward obtained from PARGS follow the true utility distribution?** As pointed out by the authors, there are multiple ways to assign the token-level rewards given the final reward. I think this assumption and the proposed loss function will favor a uniform reward distribution among tokens, rather than rewards being sparsely distributed to only a few tokens. Interestingly, [1-2] has empirically shown that rewards are sparse. For instance, in math/multiple-choice questions, when the value function is estimated by sampling 5 continuations from each prefix, the reward is usually focused on a *single* token [1]. **Question:** How is the final reward distributed to each token in PARGS (uniform vs. sparse), and does that follow/contradict [1-2]'s results?

2. **Missing tasks with verifiable rewards** Following the previous argument, a good way to justify PARGS and its underlying assumption would be testing on domains with verifiable rewards, like math, multiple-choice problems, or coding. My personal experience is that improving performance in these tasks by steering (directly manipulating token probabilities) is more challenging than tasks like toxicity or summarization. It would be nice to explore the utility of the proposed PARGS on these tasks.

[1] Bigelow, E., Holtzman, A., Tanaka, H., & Ullman, T. (2024). Forking paths in neural text generation. arXiv preprint arXiv:2412.07961.
[2] Lin, Z., Liang, T., Xu, J., Wang, X., Luo, R., Shi, C., ... & Tu, Z. (2024). Critical Tokens Matter: Token-Level Contrastive Estimation Enhence LLM's Reasoning Capability. arXiv preprint arXiv:2411.19943.

---

> ### Author Response · Authors · 2025-06-03
>
> Thank you for your review and comments.
>
> ## Does the assumption hold
>
>
> If we are given two partial sequences A and B, the one with the best extension to full sequence should be preferred. If the winning sequence from the preference dataset is close to the optimal sequence or human generated, then our assumption would do the correct thing.
>
>
> ## Tasks with verifiable rewards
>
> Our goal for this work was to conduct comprehensive experiments on summarization and dialogue, since all the RGTG baselines also presented experiments in these domains.
>
> Math,Coding and other tasks with verifiable rewards are important extensions to RGTG and we leave it for future work.

---

> > ### Comment · Reviewer_2dhF · 2025-06-03
> > **Response to the authors**
> >
> > Dear authors, thank you for your response.
> >
> > Regarding the first point, I would like to clarify two things:
> > - I completely agree that *"If we are given two partial sequences A and B, the one with the best extension to a full sequence should be preferred."*
> > - However, the following sentence: *"If the winning sequence from the preference dataset is close to the optimal sequence or human-generated, then our assumption would do the correct thing."* is a statement I cannot agree with at all. In the higher level, winning sequences should not be considered optimal, given that most preference data are sampled from the initial policy (which is not an optimal policy because it can be eventually improved by RL) and the overall losing sequence in the pair might be better in some criteria than the winning one (refer to Ultrafeedback, Prometheus 2, ...). In the lower level, the important issue is that the observed sequence is only a single path of the entire search space (exponential tree), and we cannot ensure that there exists an alternative continuation in the losing prefix that achieves higher reward than the observed winning sequence (as pointed out by reviewer Z2q2). The proposed method does not explore multiple continuations to reduce the variance of the observed maximum reward, which is the only way to convince that this assumption holds. Overall, the authors' insufficient explanation of this assumption in the rebuttal amplifies my criticism.
> >
> > Regarding the second point, I agree with the authors that demonstrating the strength in the summarization/dialogue tasks is sufficient for showcasing the empirical strength. However, since the assumption of this work is highly questionable, showing empirical gains and performing suggested analyses in reasoning tasks (with verifiable rewards) will be the best way to justify the paper's contribution.
> >
> > Given these two points, I maintain the score of 5 and increase confidence. I still think that the empirical results are promising. However, the core assumption is not being properly supported at all (both in the paper and rebuttals), even when all reviewers unanimously pointed it out as a reason to reject.

---

> > > ### Author Response · Authors · 2025-06-10
> > >
> > > Thank you for the follow-on comments and question. We agree with some of your observation and in our previous answer were presenting an ideal case for our assumption. We present experiments on the Anthropic-HH dataset, TL;DR, Ultrafeedback and human corrected Machine Translation dataset. The HH dataset typically pairs harmful and harmless sentences next to each other. Our assumption in that case is reasonable. The machine translation dataset has human corrections to erroneous translation that we use as preference data.
> > >
> > > We acknowledge that other datasets like Ultrafeedback are sampled from an initial policy and as you mention "we cannot ensure that there exists an alternative continuation in the losing prefix that achieves higher reward than the observed winning sequence". However, we do push back on the assertion that this assumption is highly questionable. We may not be able to ensure that it always holds (which we do not argue against), but it leads to a simple method, with strong empirical results and avoids some obvious pitfalls of prior works such as using a full-sequence reward model to score partial sequences. Moreover we don't require additional annotation or reliance on multiple completions from a language model.

---

> > > > ### Comment · Reviewer_2dhF · 2025-06-10
> > > > **Response to authors**
> > > >
> > > > Dear authors,
> > > >
> > > > The follow-up response is much more informative and helpful. Thank you for the effort in writing this!
> > > >
> > > > The authors' perspectives are reasonable and understandable. I agree that the empirical strength of the proposed method can be used as evidence for the validity of the assumption, which led me to give a borderline (not strong reject) score. However, I decided to maintain my score unless there is a piece of direct evidence that proves the assumption.
> > > >
> > > > I believe that a toy experiment or two will be enough to show that this assumption is true, and I strongly encourage authors to provide experimental results that back the claim. I would be happy to increase the score if I can agree that the assumption holds (for most of the time & in diverse tasks)!

---

> > > > > ### Author Response · Authors · 2025-06-11
> > > > >
> > > > > Thank you for your positive response and suggestion. We did a preliminary study on 50 samples from the TL;DR test set. We took the losing sequence, cut it at $25$% and $50$ % sequence lengths and used $\pi_{ref}$ to generate multiple completions. Next we compared the reward score of the winning sequence vs all the losing sequences. The winning sequence received a higher reward than all completions for $62$%  and $60$% of the samples respectively. Please note 2 things i) the test accuracy of the full sequence reward model is around $70$% and ii) we need to generate multiple completions for the winning sub-sequence for a fair comparison. Nevertheless, this result is encouraging. We will include a human evaluation of this scenario in the final version of the paper for a more accurate picture.

---

> > > > > > ### Comment · Reviewer_2dhF · 2025-06-11
> > > > > > **Response to authors**
> > > > > >
> > > > > > Dear authors,
> > > > > >
> > > > > > Thanks for conducting this experiment, and I am glad that my suggestion was accepted by the authors.
> > > > > >
> > > > > > I think this initial result is positive. I strongly agree with points (1) and (2), and it would be really nice to see more thorough (human eval, more cutting points, more datasets...) results in the final version.
> > > > > >
> > > > > > As authors provide experiment results that indicate the assumption in question holds reasonably often, with the hope of seeing more detailed analysis in the final draft, I increase the score. Thank you for participating in the long rebuttal!

---

### Official Review · Reviewer_rFy9 · 2025-05-08

**Rating:** 6
**Confidence:** 3
**Ethics Flag:** 1

**Summary:**

This paper investigates reward-guided text generation (RGTG) as an alternative to standard offline reinforcement learning from human feedback (RLHF). The authors identify a key limitation of tokenwise RGTG: when the reward model is trained solely on full sequences, it may assign arbitrary rewards to partial sequences, leading to inadequate guidance during autoregressive decoding. To alleviate this issue, the authors propose explicitly training the reward model on partial sequences using the Bradley-Terry (BT) model framework, thereby enabling more reliable token-level rewards during generation. The authors further provide a theoretical analysis showing that the resulting policy is proportional to the ratio of two RLHF policies over sequences of different lengths, reflecting a tradeoff between avoiding pathological behavior of the reward model on partial sequences and maintaining tractability. Experimental results on summarization and dialogue generation tasks show the effectiveness of the method compared to recent RGTG baselines.

**Questions To Authors:**

Please refer to "Reasons To Reject"

**Reasons To Accept:**

1. The paper is clear and well-structured, especially the Preliminaries section, which provides a thorough introduction to RLHF, Reward-Guided Generation, and the relevant formulas related to these concepts. This section effectively guides the reader through foundational concepts, enabling a better understanding of the subsequent analysis.
2. The mathematical derivations are well executed, with sufficient details provided throughout. Furthermore, the proofs included in the appendix are clearly presented, facilitating easy comprehension for the readers.
3. By avoiding the need for expensive fine-tuning of the base LLM, the method is computationally efficient compared to traditional RLHF methods like PPO and DPO, while achieving competitive performance.
4. The method is shown to generalize across different LLMs and datasets, highlighting its robustness and flexibility in diverse text generation contexts.

**Reasons To Reject:**

1. The theoretical discussion around the Bradley-Terry model applied to partial sequences feels underdeveloped. Although the paper explains the ratio of two distinct RLHF policies, it does not sufficiently justify why this particular approach is the best one or explore the limitations and potential drawbacks in depth. The discussion could benefit from a more thorough exploration of the implications of using a ratio of RLHF policies and how this may affect the quality of the generated text.
2. The paper contains several grammatical issues that may affect clarity. For example, in the conclusion section, the sentence “it performs better than a recent RGTG methods ...” contains a number agreement error ("a" with a plural noun), and in the related work section, the sentence “Different from our work, Zhao et al. (2024) a reward-guided decoding method ...” appears to be missing a main verb. Addressing these issues would improve the readability and overall quality of the writing.
3. There appears to be a minor issue in the reward function derived from Equation (3), as it appears that the coefficient 1/β is missing in front of . It might be helpful to verify whether this omission was intentional or an oversight in the mathematical derivation.
4. One potential weakness is that the paper assumes preference over full sequences implies preference over corresponding prefixes. However, this assumption may not hold in practice, as annotators typically do not provide preferences over partial sequences, and preferences may depend on tokens later in the sequence. This assumption may introduce bias into the learned reward model.
5.  Another potential issue is that while the paper cites Deng & Raffel (2023) in several places (e.g., in the Introduction as a related approach), it does not include this method as a baseline in the experiments. It would be helpful if the authors could clarify why this method was not included, or consider adding it if feasible.
6. The paper uses a mix of different evaluation metrics (e.g., win-tie rate, reward score), but does not provide a clear rationale for why these specific metrics were chosen or how they directly relate to real-world performance. It is unclear how well these metrics align with human expectations or the actual task requirements.

---

> ### Author Response · Authors · 2025-06-03
>
> Thank you for your detailed review and questions. We hope that our response will satisfy your concerns. Please note that we have corrected the typos and writing errors that you highlighted.
>
> ## Bradley-Terry Model on Partial Sequences
>
> We note that baseline RGTG methods such as ARGS (Khanov et, al 2024), CD (Mudgal et. al, 2024) and RAD (Deng and Raffel, 2023) do not present any connection to offline RLHF. Our analysis for PARGS shoes that we recover a ratio of two distinct RLHF policies. We argue that this approach is the best because it avoids the pitfalls of Theorem 1, and has strong (with significance testing) empirical performance.
>
> ## Missing term in Equation 3
>
> The equation has $\beta$ inside the exponential which shows up as $\frac{1}{\beta}$ in the other equations.
>
> ## Method Assumptions
>
> If we are given two partial sequences A and B, the one with the best extension to full sequence should be preferred. If the winning sequence from the preference dataset is the optimal sequence or human generated, then our assumption would do the correct thing.
>
> ## RAD Baseline
>
> RAD (Deng & Raffel, 2023) is very similar to CD-Fudge method (Mudgal et. al ,2024) in that they distill a tokenwise reward model from a full-sequence reward model using a square loss function. We included CD in all our experiments. Nevertheless, we present the results of RAD on the TLDR dataset.
>
>
>
> | Method | Reward $\pm$ Standard Error |
> | -------- | -------- |
> | Top-K Sampling     | -0.11 $\pm$ 0.28 |
> | RAD     | 0.11 $\pm$ 0.25 |
> | CD     | 0.32 $\pm$ 0.33 |
> | ARGS     | 1.57 $\pm$ 0.21 |
> | PARGS     | 2.36 $\pm$ 0.20 |
>
> We observe that PARGS has the best performance when looking at the reward score. We will add the complete results to the paper.
>
> ## Metrics
>
> We briefly discuss the different metrics that we employ starting line 298. Following Khanov et. al, 2024, we use reward models trained on full sequences (preference data) as an oracle to measure the alignment of generation with human preferences. Next we use GPT-4 as a proxy to human evaluation to rank between pairs of generations. We provide the prompt in the appendix. This has been shown to align with human preferences (Rafailov et al.,2023). We also present a human evaluation study in Appendix H.
>
> We present an additional experiment on applying PARGS on Machine Translation using  a reward model trained on post-edit dataset in Appendix F. For these experiments we look at the BLEU score. Finally, we look at generation diversity using the Rouge-L metric in appendix F.

---

> ### Comment · Reviewer_rFy9 · 2025-06-05
>
> I have read others' comments, as well as the authors' response, which have addressed most of my concerns, I maintain my original score.

---

> > ### Author Response · Authors · 2025-06-10
> >
> > Thank you for acknowledging that most of your concerns have been addressed.

---

### Decision · Program_Chairs · 2025-07-08

**Decision:**

Accept

**Comment:**

This paper proposes a solution to the problem of token-wise guided text generation, where sequence-level rewards are used to influence token-level decisions. The reviewers main pros and cons are:

**Pros:**
- The paper is clear and well-structured; the mathematical derivations are coherent, and the conceptual ideas behind the method are well articulated.
- Empirical results demonstrate that the proposed method outperforms existing RGTG baselines.

**Cons:**
- The theoretical justification is not well developed.
- There is a lack of experiments to show the generalizability of the proposed method.
- The proposed work involves a significant amount of overhead during inference.
- Some critical baseline comparisons are missing.